

# Environmental DNA simultaneously informs hydrological and biodiversity characterization of an Alpine catchment

Elvira Mächler[1, 2], Anham Salyani[3], Jean-Claude Walser[4], Annegret Larsen[3, 5], Bettina Schaefli[3, 6, 7], Florian Altermatt[1, 2], and Natalie Ceperley[3, 6, 7]

[1]Eawag: Swiss Federal Institute of Aquatic Science and Technology, Department of Aquatic Ecology, Überlandstrasse 133, CH-8600 Dübendorf, Switzerland
[2]Institute of Evolutionary Biology and Environmental Studies, University of Zurich, Winterthurerstrasse 190, CH-8057 Zürich, Switzerland
[3]Faculty of Geosciences and Environment, Institute of Earth Surface Dynamics, University of Lausanne, 1015 Lausanne, Switzerland
[4]Federal Institute of Technology (ETH), Zürich, Genetic Diversity Centre, CHN E 55 Universitätstrasse 16, 8092 Zürich, Switzerland
[5]Soil Geography and Landscape Group, Wageningen University, Droevendaalsesteeg 3, 6708 PB Wageningen, The Netherlands
[6]Geography Institute, University of Bern, 3012 Bern, Switzerland
[7]Oeschger Centre for Climate Change Research, University of Bern, Switzerland

**Correspondence:** Elvira Mächler (elvira.maechler@gmail.com) and Natalie Ceperley (natalie.ceperley@giub.unibe.ch)

**Abstract.** Alpine streams are particularly valuable for downstream water resources and of high ecological relevance, however a detailed understanding of water storage and release in such heterogeneous environments is still often lacking. Observations of naturally occurring tracers, such as stable isotopes of water or electrical conductivity, are frequently used to track and explain hydrological patterns and processes. Importantly, some of these hydrological processes also create microhabitat variations in

Alpine aquatic systems, each inhabited by characteristic organismal communities. The inclusion of such ecological diversity in a hydrologic assessment of an Alpine system may improve our understanding of hydrologic flows while also delivering biological information. Recently, the application of environmental DNA (eDNA) to assess biological diversity in water and connected habitats has gained popularity in the field of aquatic ecology. A few of these studies have started to link aquatic diversity with hydrologic processes, but hitherto never in an Alpine system. Here, we collected water from an Alpine catch-

ment in Switzerland and compared the genetic information of eukaryotic organisms conveyed by eDNA with the hydrologic information conveyed by naturally-occurring, hydrologic tracers. Between March and September 2017, we sampled water at multiple time points at 10 sites distributed over the 13.4 km$^2$ Vallon de Nant catchment (Switzerland). The sites corresponded to three different water types and habitats, namely low flow or ephemeral tributaries, groundwater fed springs, and the main channel receiving water from both previous mentioned water types.

Accompanying observations of typical physico-chemical hydrologic characteristics with eDNA revealed that in the main channel and in the tributaries the biological richness increases according to change in streamflow, *dq/dt*. Whereas, in contrast, the richness in springs increased in correlation with electrical conductivity. At the catchment scale, our results suggest that transport of additional, and probably terrestrial, DNA into water storage or flow compartments occurs with increasing





streamflow. Such processes include overbank flow, stream network expansion, and hyporheic exchange. In general, our results highlight the importance of considering the at-site sampling habitat in combination with upstream connected habitats to understand how streams integrate eDNA over a catchment and to interpret spatially distributed eDNA samples, both for hydrologic and biodiversity assessments. At the intersection of two disciplines, our study provides complementary knowledge gains and

identifies the next steps to be addressed for using eDNA to achieve complementary insights into Alpine water sources. Finally, we provide recommendations for future observation of eDNA in Alpine stream ecosystems.

**Keywords:** snow melt, groundwater, Alpine, eDNA, hydrology, metabarcoding

# 1 Introduction

Alpine environments are often considered to be water towers for lowland areas (Viviroli et al., 2007), hotspots for biodiversity (Körner, 2002), and will likely be disproportionately affected by climate change compared to other areas (Jacobsen et al., 2012; Grabherr, 2009). The complex topography of mountain areas results in a tremendous variation in the physical environment in terms of elevation, slope, and exposure, which subsequently creates large gradients of received radiation and thus extreme variations in temperature at small scales. Subsequently, the topography and temperature gradients translate into a landscape with

an enhanced heterogeneity of potential water flow paths. These paths are additionally governed by highly seasonal incoming water (rain or snow), which is released as a seasonal alternation of runoff from rain, stored snow, and snowmelt. Consequently, channel networks in Alpine areas are known to regularly expand and contract, leaving a portion of streams ephemeral (Godsey and Kirchner, 2014; van Meerveld et al., 2019).

Despite the importance of mountain water resources, hydrologic processes are still often poorly characterized in mountain

catchments. From a hydrologic as well as from an ecological viewpoint, the central phenomenon determining habitat availability is the origin of stream water, for example, whether streams are fed by seasonal snow melt or by recent precipitation (Nolin, 2012; Cochand et al., 2019; McGuire and McDonnell, 2015). In head-waters of mountain environments, the relevant hydrologic mechanisms are difficult to observe because they occur out of direct reach, for example in the atmosphere or the subsurface (McDonnell et al., 2007). There is an ongoing search for hydrologic tracers to decompose river water into its most recent stor-

age compartment (e.g., glacier, snow, soil, groundwater) and to identify when and where it entered the stream network (Abbott et al., 2016; Blume and Van Meerveld, 2015; Williams et al., 2009; Mosquera et al., 2018).

Spatial and temporal variations in the physico-chemical properties of water in Alpine catchments, such as in temperature, discharge, stable isotopes, and electrical conductivity (E.C.), offer clues to identify water flow from glacier, snow, soil and groundwater and its associated time scales (Zuecco et al., 2019; Tetzlaff et al., 2015; Penna et al., 2014; Klaus and McDonnell,

2013; Beria et al., 2018). For example, E.C. can relatively consistently discriminate snowmelt, rain, and potentially glacier melt (all of them having very low levels of sodium) from groundwater (which typically has much higher levels of solutes)





(Williams et al., 2006; Cochand et al., 2019; Kobierska et al., 2015). On the other hand, water temperature cannot be used as a conservative tracer, because it quickly changes along a water course, but can provide insights to separate surface water (having a strong diel variation) and influxes of water from the subsurface (having more constant temperatures; Constantz, 1998; Comola et al., 2015; Hoehn and Cirpka, 2006; Westhoff et al., 2007; Selker et al., 2006). Finally, stable isotopes of water, $\delta^{18}$O and

$\delta^2$H, have classically been used as tracers of source water, flow paths, and precipitation contributions. Precipitation becomes depleted, or has lower values when it falls at higher elevations and in lower temperatures, especially when it falls as snow. This natural variation in isotopic ratio in precipitation according to season and elevation offers clues to when and where it entered the system. However, in high Alpine contexts, it often falls short in offering additional insights into dominant flow paths because the isotopic ratio of all potential water sources is dominated by the precipitation phase of incoming water (rain versus snow;

Lyon et al. 2018; Beria et al. 2018; Penna et al. 2018). Landwehr and Coplen (2006) improved upon the direct observation of isotopes with the line-conditioned excess (lc-ex), a metric that is equal to the residual between two isotopes of water, $\delta^{18}$O and $\delta^2$H. Higher values of lc-ex, thus, indicate when water has experienced losses to evaporation or other fractionation, allowing it to be used as an additional tracer alongside the ratios of oxygen and hydrogen isotopes individually.

Similar to their high variations in water proprieties, Alpine catchments are typically dominated by high fluctuations in

discharge and corresponding changes in the extent of the stream network (Godsey and Kirchner, 2014). The change in discharge at a catchment's outlet over time, *dq/dt*, can thus serve as a proxy for stream network recession and expansion (Biswal and Marani, 2010; Mutzner et al., 2013). When *dq/dt* is negative, it indicates how the upstream stream network contracts and the rate at which stored water, whether in soil or as groundwater, continues to flow into the stream). When it is positive, it indicates network expansion after a precipitation input event, whether rainfall or snow melt, which has lead to subsurface saturation,

overland flow and fast subsurface flow.

In the highly variable Alpine environment, water flow and its properties can mean proliferation or eradication of habitats thus determines persistence of biodiversity. Purely disciplinary views to address current global changes and the consequences for habitats and diversity are not sufficient to address this challenge. Thus, especially in the context of global changes, it has become clear that purely disciplinary views are not sufficient to address some of the current environmental challenges. Future indicators

that simultaneously offer information about the hydrological and ecological dimensions of the environment may be critical to get a coherent and complete picture of aquatic systems. Such novel methods should complement existing tools by either improving their accuracy, or by revealing additional, complementary insights into a system, such as complementing hydrologic information with biological information. Alpine aquatic systems are especially suited for such an interdisciplinary approach, because the high variability and complexity of physio-chemical properties and dynamic flowpaths is known to be matched

by equally complex and diverse biological habitats that are inhabited by highly specialised organismal communities (Brown et al., 2003; Milner and Petts, 1994; Ward, 1994). For example, midges of the genus *Diamesa* are characteristic or sometimes even the only inhabitants of rivers dominated by glacial input, resisting the cold temperatures and high turbidity (Milner and Petts, 1994). Contrarily, plecopterans of the genus *Protonemoura* or the species *Leuctra nigra* are typical inhabitants of Alpine springs (Staudacher and Füreder, 2007; Hahn, 2000). The same is true for microorganisms, where higher abundances of $\alpha$-

Proteobacteria were found in glacial streams than in other aquatic systems, likely reflecting the trophic status (i.e., the primary





productivity) of the habitat (Battin et al., 2004). Finally, algae of the genus *Chamaesiphon* and the species *Hydrurus foetidus* are indicative of glacier-dominated sites (Hieber et al., 2001). Consequently, the presence and drift of biological organisms are expected are not only of high ecological relevance, but also have the potential to trace connectivity of the stream network, a hypothesis which has been confirmed by some studies for bacteria (Pfister et al., 2009; Lan et al., 2019) and diatoms (Wang

et al., 2017).

In recent years, the use of environmental DNA (eDNA) has been shown to be highly powerful for monitoring organisms in aquatic ecosystems (Valentini et al., 2016; Deiner et al., 2015; Bohmann et al., 2014; Altermatt et al., 2020). All organisms leave traces of their genetic material in the environment, and their DNA can be collected in environmental samples. With a single sampling technique, eDNA samples can detect communities of a broad taxonomic scale by using a high through-put sequencing

approach of a barcoding region, so called metabarcoding. In an ideal case, the barcoding region allows identification at the level of species. In stream systems, several studies have shown that eDNA can travel for distances of 10 to 100 of kilometers (e.g., Deiner and Altermatt 2014; Pont et al. 2018). Thus, stream networks congregate eDNA from upstream areas (Deiner et al., 2016), highlighting its potential to derive flow path information at the catchment-level as well as reconstructing diversity patterns using hydrologic models (Carraro et al., 2020a). Recent work from lowland streams and their waste-water inflows also

showed that mixing of different water sources can be traced to a high level of reliability with eDNA (Mansfeldt et al., 2020). While artificially introduced DNA attached to particles has already been used as a hydrologic tracer (Dahlke et al., 2015; McNew et al., 2018; Foppen et al., 2013), naturally occurring DNA has just started to enter into the repertoire of hydrologic tracers (Good et al., 2018; Carraro et al., 2018, 2020a, b). Particularly, it may complement existing physical tracers by not only indicating possible water sources, but also indicating their organismal communities, which are regularly used as a water quality

component.

In this study, we tested whether the inclusion of eDNA in a hydrologic assessment of an Alpine system improves our understanding of hydrologic flows while simultaneously delivering biological information. We also explored how this would benefit biologists seeking to quantify Alpine diversity by providing clear recommendations regarding where and when to sample eDNA in river networks for assessments of diversity. To do so, we repeatedly sampled water in an Alpine catchment from spring to

summer. We selected ten sites corresponding to three different water types: low flow streams that are likely ephemeral, fed by snow melt, glacier melt or rain (tributaries); groundwater emerging from subsurface (springs); and the higher flow stream fed by the other two (main channel). Throughout this study, we will refer to these three water types to encompass the characteristic flows and sources of this catchment and their corresponding aquatic habitats. Firstly, we hypothesized that different Alpine water types carry significantly different eDNA signals that can be used to discriminate between them. Secondly, we hypothesized

that hydrologic variability, i.e. the change in stream- and subsurface water flow as observed through various physico-chemical indicators, drives the temporal and spatial eDNA signal. Thirdly, we expected the sum of eDNA from upstream sampling points to shape the eDNA signal at the most downstream catchment outlet. Our study provides an initial assessment of the spatial and temporal heterogeneity of eDNA in an Alpine stream system. We give concrete recommendations for sampling eDNA in this context and identify opportunities but also challenges to its possible use for gaining additional hydrologic insights.





## 2   Methods

We monitored eDNA in an Alpine catchment in Switzerland in parallel with hydro-meteorologic observations. On 11 field days in 2017 (for exact dates see Table S1), we sampled eDNA and simultaneously observed the E.C., water temperature, and stable isotope ratios of the water ($\delta^{18}$O, $\delta^2$H). In addition, we continuously monitored discharge at the catchment outlet

and meteorological parameters at 4 stations distributed across the catchment (Fig. 1). We first describe the field site and instrumentation for hydrologic observations. Then, we explain the eDNA sampling and laboratory procedures before describing the data analysis and used statistics.

### 2.1   Site description

The *Avançon* is the main stream in the *Vallon de Nant* catchment (Canton of Vaud, Switzerland, Fig. 1). The 13.4 km$^2$ catchment

ranges from 1200 m a.s.l. (as defined by a gauging station at 46.25301$^o$N, 7.10954$^o$E) to 3051 m a.s.l. (*Le Muveran*). It has a dominant north exposition, with almost no direct sun during winter in its upper parts, allowing a small glacier (*Glacier de Martinets*, 0.36 km$^2$, GLAMOS 1881-2018) to persist at relatively low elevation. Permafrost is likely present above 2400 m a.s.l. (Giaccone et al., 2019). The streamflow regime is of the nivo-pluvial type (Aschwanden and Weingartner, 1985), with a monthly streamflow peak in June. The stream network in the catchment consists of streams that have been classified as

first, second, and fourth order streams according to the Strahler terminology. Correspondingly they have been characterized as having low (< 0.05 m$^3$/s and occasionally ephemeral) and medium annual mean discharge (0.05 - 1 m$^3$/s), which we will refer to as tributaries and main channel, respectively, see Schaffner et al. (2013); Pfaundler (2005), and Table S2 in the Supplementary Material. The stream network is fed by melt from the seasonal snow pack including avalanche deposits, which can persist into the summer months, and by drainage from the soil (abundant unconsolidated alluvium) and from deeper groundwater via

springs. Permafrost and glacier melt enters the catchment in the upper tributaries.

### 2.2   Hydrologic parameters

#### 2.2.1   Instrumentation and field observation

Water level at the outlet (Fig. 1) was measured using a VEGAPULS WL-61 optical height gauge (VEGA, Schiltach, Germany) over a geophone weir and discharge was determined using an established rating curve (Ceperley et al., 2018). A network of four

distributed meteorological stations (at 1253, 1500, 1780, and 2100 m a.s.l., Fig. 1) included precipitation and air temperature, mainly measured with Lufft-WS 300/400 (Fellbach, Germany), solar radiation with Apogee incoming radiometer (SP230, Logan, Utah, U.S.A.), mounted in an energetically autonomous wireless network with a GPRS connection (Sensorscope Sarl, Lausanne, Switzerland, Michelon 2017; Ingelrest et al. 2010). At a single point (1253 m a.s.l.), a MADD tipping bucket rain gauge provided back up measurements (Fallot 2013, Yverdon, Switzerland). Data from the Swiss automatic meteorological

station network (MeteoSwiss, 2019) is used to fill gaps of the local observations (12 missing days in 2017). Water temperature was recorded every 15 minutes or more frequently at all but one sampling site (TT, see Fig. 1) with Hobo temperature/light



**Figure 1.** Map of *Vallon de Nant* showing sampling sites and stream network. Main channel sites are red squares, tributaries by green triangles, and springs are represented with blue circles, in all three cases, shading lightens with elevation. The two letter codes are used to identify the sites (see also Table S2). The main channel is the *Avançon de Nant*, which is shown in dark cyan. Predominately ephemeral tributaries of various origins are shown in turquoise. The remaining *Glacier de Martinet* is shown in light blue. Yellow stars with a red dot in their center indicate where meteorological stations also corresponded to points of rain collection for isotope analysis. Discharge observation occurred at the outlet which is indicated by the MR/ER sampling site icon. The location of the catchment on an outline of Switzerland with its main hydrologic network is shown to the lower right. On the right are pictures of the main channel (MR/ER, top), a tributary (ST, middle) and a spring (BS, bottom). Colored circles show sampling points and arrows show direction of water flow.





pendants and one temperature/conductivity logger for parts of the study period (Supplementary Material Table S2, Onset Computer Corporation, Bourne, MA, USA). One WTW-Tetracon 325 (Xylem Analytics, Germany) was installed, immersed at the outlet for the entire year and logged temperature every minute.

At each point, temperature and E.C. were measured simultaneously with water sampling using a WTW probe (multi-3510 with a IDS-tetracon-925, Xylem Analytics, Germany). When temperature was not measured during sampling, it was substituted by an observation from the continuous water temperature loggers. When E.C. was not measured in the field, it was post-analyzed using a glass 6 mm probe in the laboratory (Jenway 4510, Staffordshire, UK).

A total of 135 composite rain samples were collected using funnels flowing into insulated bags at 3 locations corresponding to the rain gauges (1253, 1500, and 2100 m a.s.l., see Fig. 1) and emptied approximately weekly or biweekly during the summer seasons between June 2016 and November 2018 (3). In total between February 2016 and April 2018, 199 snow samples were collected in a distributed manner across the catchment, with between 3 and 26 samples per month in the winter, and included the entire snow pack depth when possible (see additional details in Table S3).

Snow covered area was calculated as a ratio from 0 to 1 over the non-forested part of the catchment using 21 Landsat 8 images and 24 Sentinel 2 images between August 2016 and December 2017 (Michelon et al., 2018). Time steps between the images ranged from 0 to 43 days, averaging 11 days. A linear interpolation of the combined time series was used to provide a fractional area covered with snow per sampling date. Satellite images were taken on average 7 days before or after sampling, though some were on sampling days and others were up to 20 days prior or after.

### 2.2.2 Flow calculations

Both baseflow and $dq/dt$ were calculated from the discharge data at the outlet at the highest recorded streamflow resolution (10 minutes). Baseflow was defined as the lowest value in a ten day moving window. On the other hand, the change in discharge over time, $dq/dt$, was determined as the slope of the linear regression fitted on a 48-hour window preceding the moment that each sample was collected. This two-day window was chosen to reduce sensitivity to the noise of streamflow records, but it also corresponds to the persistence of eDNA in the environment while also preventing the domination of the diurnal streamflow cycle. For this study, $dq/dt$ is a proxy for how the stream network contracts and expands in combination with overland flow. Rather than assuming a particular scaling relationship, we took the $dq/dt$ at the outlet to be a good proxy for the upstream network contraction, which is justifiable as significant and rapid increases in discharge correspond to precipitation events covering large parts of the catchment (i.e., are not concentrated on some small sub-catchment) in the *Vallon de Nant* (Michelon et al., 2020). Detailed results of a sensitivity analysis regarding the optimal time window length are reported in the Supplementary Material (Fig. S1).

### 2.2.3 Stable isotopes of water

Water was sampled at the 10 stream sites directly following the sampling of eDNA using 12 mL amber screw vials with a solid polypropylene cap and a silicone rubber/PTFE septa (BGB Analytik, Boeckten, Switzerland) at the sampling points (Fig. 1 and Supplementary Material Table S3). Water was analyzed for stable isotope compositions of oxygen and hydrogen using a





Picarro Wavelength-Scanned Cavity Ring Down Spectrometer (WS-CRDS) 2140-i (Santa Clara, California, U.S.A.). Samples were injected a minimum of six times and the last three measurements were averaged to calculate a raw value. Values were corrected according to a standard curve determined with internal standards, which are regularly calibrated against international standards VSMOW (Vienna Standard Mean Ocean Water) and SLAP (Standard Light Antarctic Precipitation) of the IAEA

(International Atomic Energy Agency). Delta units of isotope compositions are reported (Coplen, 1994).

All available samples of precipitation were used to determine a local meteoric water line (LMWL, Fig. S2), which was used to determine the line-conditioned excess (lc-ex, $L$), or offset from the LMWL:

$$L = \delta^2 H - 7.82. * \delta^{18}O - 10.47. \tag{1}$$

A positive value of lc-ex falls above the LWML and a negative one below (Landwehr and Coplen, 2006). The value of lc-ex

reflects a combination of the source water and the physical processes that have occurred since precipitation, with more negative values indicating more evaporation and condensation (Sprenger et al., 2016).

## 2.3   environmental DNA

### 2.3.1   Sampling and laboratory procedure

eDNA was sampled by filtering 250 mL of stream water on each of four filter replicates per site using GF/F filters (25 mm

diameter, 0.7 μm, Whatman International Ltd., Maidstone, UK) directly in the field at the 10 sites immediately before stable isotopes were sampled. Following filtration, we dried the filter by squeezing air through it, rolled the filter with tweezers, and stored the filter into a 1.5 mL tube following the description of Mächler et al. (2018). If it was not possible to collect 250 mL on a single filter, we used more filters until we reached the goal of 1 L filtered water per site. The tubes were transported on ice and were then stored at –20 °C. We implemented a negative filter control consisting of 1 L MilliQ water that was previously

treated with UVC light, before beginning sample collection in the field, to verify a DNA-free status of used materials. Sites were sampled in all cases except during extreme avalanche conditions, snow cover of the sampling site, or when the sampling site was dry.

We extracted the DNA from the filters after all samples had been collected. We used the DNeasy® Blood and Tissue kit (Qiagen, Hilden, Germany) following the protocol for animal tissue besides a few changes (see Supplementary Material,

Section S3.1). In order to reduce biases due to lab procedure, we extracted the samples in a random order. During extraction, we combined two of the four filter replicates, resulting in two extractions per sampling event at the individual sites. We included a negative extraction control, containing a previously with UVC light treated filter, for each batch of extractions, resulting in 8 extraction controls. The extracted DNA was stored at –20 °C until further processing.

Generally, eDNA exists in low concentrations in the environment. Thus, to sequence the DNA region of interest, it first

needs to be amplified. In our study, we targeted a subsection of a previously identified barcoding gene region for eukaryote species (Hebert et al., 2003), a 313 base pair fragment of the cytochrome oxidase I gene (COI, Geller et al. 2013; Leray et al. 2013, Table S4). We used the Illumina dual-barcoded two-step protocol to amplify and sequence our samples consisting of





polymerase chain reactions (PCR), replicates to minimize stochasticity effects, implementation of additional controls, indexing, and pooling (see Supplementary Material, Section S3.2 for individual steps).

eDNA laboratory methods are optimized for the detection of low amounts of DNA and thus very sensitive to contamination. In order to reduce false-positives, we followed the previously described laboratory precaution for work with eDNA (Deiner

and Altermatt, 2014; Deiner et al., 2015; Mächler et al., 2015). Reused field material, such as filter housings and syringes, was soaked 40 minutes in 2.5% sodium hypochlorite (i.e., bleach), rinsed with deionized water and treated with UVC light prior to the reuse in the field. Furthermore, four types of controls were sampled during the field (filtration) and laboratory procedure (extraction, PCR negative and PCR positive, see Supplementary Material, Section S3.3), which were run alongside the eDNA samples during the laboratory workflow and are later used for data cleaning.

### 2.3.2    Molecular data processing

The main goal of the bioinformatic analysis is to remove errors due to sequencing techniques and regroup sequences into operational taxonomic units (ZOTUs). High through-put techniques produce errors during sequencing and in order to increase appropriate sequence identification, the resulting reads are cleaned and undergo an initial check for quality (see Supplementary Material, Section S3.4). We determined variations of the amplified DNA sequences and reduced the influence of sequencing

errors with UNOISE3 (usearch v10.0.240, Edgar 2016). The detected sequence variants were clustered additionally at 99 % sequence identity to reduce diversity and to account for possible amplification errors. This clustering of the sequences is resulting in so called ZOTUs (i.e., zero-radius operational taxonomic units). In this sense, a ZOTU is a cluster of DNA sequences that are very similar and can be seen as a rough proxy of a species. As a final step, the ZOTUs were assigned to a taxonomic name if possible by comparing it to taxonomic databases(blast 2.3.0 and usearch v10.0.240, tax filter = 0.9).

However, databases are still highly incomplete (Weigand et al., 2019), and thus matches cannot be expected at a high rate.

The three types of negative controls produced during the field and laboratory procedure were used to remove noise caused by possible contamination. In a first step, we addressed possible contamination following Evans et al. (2017): For each ZOTU found in one of our negative controls, we calculated a relative frequency by dividing the sum of reads for an individual ZOTU by the sum of all ZOTUs in the negative controls, which was then used as a threshold to clean the field samples (see Supplementary

File S5). Any ZOTU with a frequency below the threshold was removed from further analysis. We then merged the two eDNA replicates for each sampling point and removed any samples that were below 20,000 reads (2 out of 107 samples), likely originating from errors in the field or laboratory work.

A ZOTU is a cluster of very similar DNA sequences and a single ZOTU can be seen as a rough proxy for a species. The number of species, also called *richness*, is a simple measure of biodiversity in ecology. However, richness is strongly affected

by the sampling effort (i.e., the higher the sampling effort, the more species that will be found). Therefore, in order to compare diversity measures between sites, data were *rarefied*, a method to standardize effort (Simberloff, 1978). The number of reads per sample is used as a proxy for sampling effort. Consequently, we rarefied the data to the sequencing depth, or number of reads, of the lowest field sample (26,759 reads). The Illumina MiSeq run resulted in 12.5 M reads and 9,858 ZOTUs after the





bioinformatics pipeline, of which 2.9 M and 9,635 ZOTUs remained after removal of contamination and rarefaction (see Table S5 in the Supplementary Material for detailed information).

## 2.4    Data analysis and statistics

We used multivariate analyses to answer three specific questions: i) whether the specific composition of eDNA (section 2.4.1)
could differentiate the three water types; ii) whether hydrologic variability (according to fore-mentioned indicators) correlated with the eDNA richness (section 2.4.2); and iii) whether the main channel's eDNA variation could be used to separate the flow contribution of springs and tributaries (section 2.4.3). All statistical analyses following the initial test were performed using R (R Core Team (2018), version 3.5.3) and the R package *phyloseq* (McMurdie and Holmes 2013, version 1.24.2).

### 2.4.1    Differentiation of water type by eDNA

In order to test the capacity of eDNA to provide insight into the flow path of the water, we assessed whether the eDNA composition varied according to sampled water type. In the study catchment, we identified three characteristic aquatic environments (tributaries, springs, and the main channel) representing unique habitats each with corresponding eukaryotic communities. We first used a Kruskal-Wallis test to compare the mean ranks of six variables ($\delta^{18}O$, $\delta^2H$, lc-ex, electrical conductivity, water temperature, and rarefied ZOTU richness) to determine if there were statistically significant differences in richness between
the seasonally spread sampling days of the different water types: main channel (M), springs (S), and tributary (T) (Kruskal and Wallis, 1952). Second, we tested whether we can distinguish eDNA samples from the three water types based on the identity of the detected eDNA sequences (i.e., ZOTUs sequence). For this analysis, we used a non-metric multidimensional scaling (NMDS, Kruskal 1964a) approach, an ordination method that represents pairwise comparisons, or rank orders, of sampling sites in a two-dimensional space. Sites clustered close together are more similar in terms of their ZOTUs than sites further
apart, which enabled us to identify if communities detected with eDNA cluster according to water type or hydrologic parameters. We used weak ties (stress type 1), allowing to break ties where equal observed dissimilarities exist (Kruskal, 1964a). We calculated the stress of the NMDS after Kruskal (1964b) as a measure of how well the configuration matches the observed data. The NMDS was plotted for the individual sampling days separately, to observe the development of differences over the sampling season.

Dissimilarity measures used for the NMDS are based on the sequence identities of the ZOTUs and were calculated to do a pairwise comparison between sites. We used an unweighted UniFrac (Lozupone and Knight, 2005) as a dissimilarity input to the NDMS, which is based on presence/absence and the phylogenetic tree (i.e., the evolutionary relationships among ZOTUs based on their sequence similarity) of the ZOTUs. We further fitted variables of interest (elevation, E.C., water temperature, total daily solar radiation, baseflow, daily snow cover area, *dq/dt*, $\delta^{18}O$, and lc-ex) onto the NMDS ordination two-dimensional
space with a commonly used method that maximized correlation between the ordination and the corresponding environmental variables (function *envfit*, R package *vegan*, Oksanen et al. 2007, version 2.5-4), to identify the directions towards which the vectors change most rapidly in the presented ordination space. Next, we focused on ZOTUs that have an taxonomic assignment





and used the R-package *indicspecies* to identify candidate taxa that are statistical significantly associated with a particular water type or combinations of those by using a permutation test (De Caceres et al., 2016).

### 2.4.2  Influence of hydrologic variability on eDNA diversity

Due to the spatio-temporal variability in water sources, Alpine catchments are a mosaic of different water types and habitats,
indicated by a high range of physical and chemical processes (Ward et al., 1999; Tockner et al., 2000). We explored the influence of water source contributions on eDNA sampled over a season associated to two variables: the change in discharge at the catchment outlet, *dq/dt* and subsurface exchange as revealed by changes in electrical conductivity (E.C.) in springs (Laudon, 1997). We used rarefied ZOTU richness, calculated for each individual sampling site and time point as a simple measure of eDNA diversity. We utilized a generalized linear mixed-effect model (GLMM, function *glmer* in the R package *lme4*, Bates
et al. 2007) with ZOTU richness (*R*) as the explanatory variable, change in discharge (*D = dq/dt*) and the water type (*W*) as fixed effects and the sampling site (*S*) as a random effect. The univariate response variable *Y=R* was modeled with vectors *X=[S]* and *Z=[D, W, D W]* of the explanatory variables associated with the fixed effect (*Z*) and the random effect (*X*). We assumed a Poisson distribution and the conditional mean $E(Y|X,Z) = \mu = g^{-1}(\eta)$ related to the linear predictor by a log-link function with inverse $h = g^{-1}$ (Crawley, 2012; Coxe et al., 2009). The identity of the sampling sites was defined as a random
effect in an intercept model, to take by-sampling site variability into account.

We compared the models with and without interaction of the two explanatory variables (*D, W*) and selected the best one based on a $\chi^2$-test, classically used to compare model fits. Since change in discharge, *dq/dt*, was negatively correlated with ZOTU richness in springs, we performed a second analysis where we replaced *dq/dt* with E.C. (*E*), in the above model. E.C. is an indicator of the subsurface exchange that defines the water type of springs likely better than *dq/dt*. E.C. was scaled and
centered for the model due to large eigenvalue ratios.

### 2.4.3  Separation of upstream water contribution

In order to examine the contribution of ZOTUs from upstream springs and tributaries to the richness in the main channel downstream, we calculated overlap in ZOTUs. We identified ZOTUs that were unique to samples from one water source, shared by only two, or all three water sources. If a ZOTU was only found in spring samples or in spring and main channel
samples, but never in any tributary samples over the whole data set, the ZOTU origin was assigned to springs. If only found in tributaries or in tributaries and main channel samples but never in any spring sites, its origin was assigned to tributaries. The ZOTU's presence in the main channel did not influence this decision, since we assumed the main channel contains eDNA of both tributaries and springs, either because it may be shared due to the similar habitat or due to downstream transport of eDNA (Deiner and Altermatt, 2014; Pont et al., 2018).
Next, we partitioned the ZOTUs in main channel samples into those originating from springs and tributaries, providing a count of how much springs and tributaries, respectively, are contributing to the main channel. We then plotted the relative contributions separately against *dq/dt* and E.C., as these were identified as major drivers in our prior analysis of ZOTU richness (see Sections 2.4.1 and 2.4.2). To confirm this relationship, we computed a generalized linear model (GLM) with the relative





contribution ($C$) as a response variable, *dq/dt (D)* and the origin of the ZOTU ($O$) as explanatory variables:

$$C = \beta_0 + \beta_1 \cdot D + \beta_2 \cdot O + \epsilon \tag{2}$$

where $\epsilon$ comes from a quasi-binomial error distribution. This choice results from i) binomial error distributions are appropriate for proportional variables; ii) our data set shows overdispersion, i.e., higher variability than what can be explained by the standard binomial error distribution. We compared the models with and without interaction of the two explanatory variables and selected the best model with a F-test. As before, we also tested a model with E.C. ($E$) as explanatory variable replacing $D$, as it might be more important to recover the input of springs:

$$C = \beta_0 + \beta_1 \cdot E + \beta_2 \cdot O + \epsilon \tag{3}$$

## 3 Results

### 3.1 Confirmation of water typology

Annual precipitation in 2017 was 1760 mm, which is close to the long term average of 1920 mm over the period 1961 - 2017 (minimum 1470 mm, maximum 2600 mm, MeteoSwiss 2011). Snow melt peaked in early June, snow cover was reduced to 9% of the watershed area by July 1 (DOY 181), and reached 0% on August 18 (DOY 229). Further details regarding the hydro-meteorological observations can be found in the Supplementary Material Section S4. In general, the three water types (main channel, tributaries, and springs) are significantly different from each other according to their physio-chemical characteristics. All comparisons of water temperature were significant ($p < 0.05$) in mean rank difference according to the Kruskal-Wallis test (Table S6), making it the best discriminator of the variables besides eDNA (Table S7). Springs had the lowest and most stable water temperature; tributaries, including the uppermost point on the main channel, UR, showed high water temperature variability, and were considerably warmer than springs; and the main channel had intermediary values. Overall, mean E.C. values were higher in springs than other observed water types, and generally higher at lower elevations. The main channel showed mid E.C. values and a narrow temporal range of E.C. values, resulting from a mixing of sources (tributaries and springs) throughout the season. Two of the between water type comparisons of E.C. were significant ($p < 0.05$) in mean rank difference according to the Kruskal-Wallis test: main channel to spring (M-S) and spring to tributary (S-T), see Table S6. Values of isotopes in the main channel were not consistently intermediary to those in springs and in the tributaries. We found significance ($p < 0.05$) in mean rank difference of $\delta^{18}O$ between main channel and spring (M-S) and main channel and tributary (M-T) but not between spring and tributary (S-T). We found significance in mean difference of $\delta^2 H$ also in those two comparisons, M-S and M-T; and in one comparison of lc-ex, M-S.

Given the seasonality of the physical processes, the variables fluctuated over the observation period, though statistical confirmation of persistent seasonality was not possible with a single season of data. Enrichment and depletion of the stable isotope concentrations in streamflow is mainly determined by contributions of snow (including snowmelt) versus rainfall (Fig. 2). The alternation of sources is visible as a large range in isotopic ratios (Fig. S3) and is particularly apparent for two of the tributaries (ST and TT), and to a certain extent for the sampling point UR in the main channel.





The analysis of the physico-chemical tracers confirmed the hypothesis underlying the sampling design, namely that the different sites correspond to different water types with typical seasonal patterns of contributions, even if there was some overlap due to accumulations of different processes. More explicitly, these sites had different contributions from rain and snow inputs at different points in time, as evidenced by stable isotopes of water, on one hand, and various ratios of surface to subsurface flow paths, on the other hand, as indicated by E.C. and temperature. Thus, each site had a unique mix of water sources and subsurface contributions and some sites had characteristics of two types. For example, the uppermost point on the river, UR, seemed to correspond to the E.C. and temperature of the main channel sometimes but at other times to those of tributaries. Likewise the most upstream tributary, UT, seemed to have an E.C. approaching that of springs compared to the other tributaries, and the E.C. of one spring, BS, is consistently lower than that of the other springs (Fig. S3). Accordingly, we expect the seasonal and spatial biodiversity patterns observed at these sites to be at least partly explained by these different origins and water flow paths, and in turn, that eDNA observations might convey information on hydrologic processes.

### 3.2 Differentiation of water type by eDNA

There was a higher variability in the number of ZOTUs (i.e., the richness) found in main channel and tributary sites (i.e., the standard deviation was higher), while sites in springs were more similar (Fig. 3, Fig. S3). All comparisons of rarefied ZOTU richness, M-S, M-T, and S-T, were significant ($p < 0.05$) in mean rank difference according to the Kruskal-Wallis test (Table S6).

Next, we focused on the analysis that included the sequence identity of the ZOTUs. The NMDS plot, which uses pairwise comparisons of multidimensional data to represent data in two dimensional space, resulted in a stress value of 0.132, indicating that the reduced dimensions represent the data fairly well (Kruskal, 1964a). While spring sites clustered slightly apart from main channel sites, this was not the case for tributary sites that are located at the intersection between main channel and springs (Fig. 4), indicating that the specific ZOTUs of the main channel are phylogenetic closer to the tributary-specific ZOTUs. The arrows in Fig. 4 indicate the direction of the fastest change in the environmental variables of E.C., $\delta^{18}$O, water temperature and lc-ex, respectively, which were all significant (Table S8), and their length is proportional to their correlation with the ordination of the genetic data (Oksanen et al., 2007). E.C. is the most important explanatory variable (i.e., it has the longest arrow in Fig. 4), followed by water temperature and lc-ex, and finally $\delta^{18}$O. The NMDS plots separated by sampling day (Fig. S5) show that in May and June (DOY 136 and 164), both tributaries and the main channel are more similar to each other compared to later in the season. The similarly low isotope values across all sites ($\delta^{18}$O =11.9 ‰, range 1.4 ‰ on DOY 164) allow us to attribute early season homogeneity in eDNA to the dominance of snowmelt, which flushes the system and reduces inter-site variability. In contrast, in late season (DOY 220), there was the highest range of water isotope values ($\delta^{18}$O, 3.05 ‰) across all sites (Fig. 3), suggesting more mixing of water originating as different types of precipitation and subsequent different trajectories of storage before being released as stream flow. However, the persistent clustering in eDNA suggests that the flood event in early August (see Fig. S5, DOY 215) equalized the detected eDNA community of tributaries and the main channel for this day, while it was spread wider the sampling days before (DOY 207) and after (DOY 220).

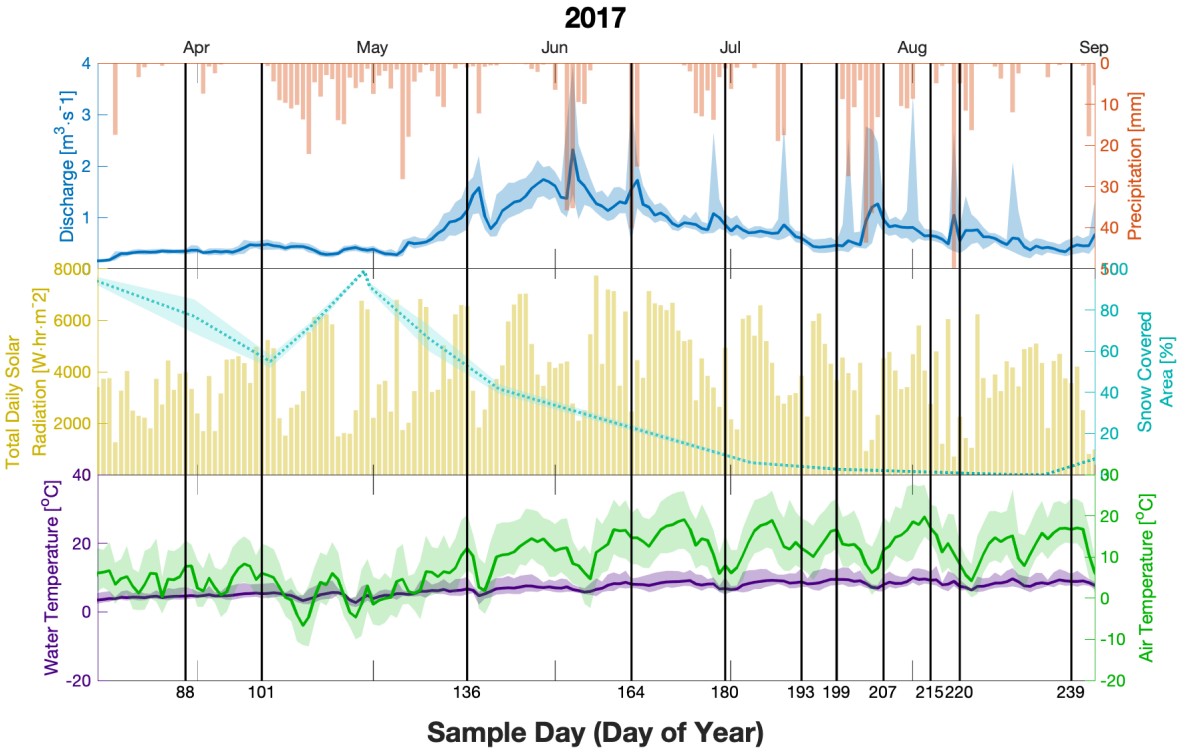

**Figure 2.** Discharge (blue, upper), precipitation (red upper), total solar radiation (yellow, middle), snow cover area (cyan, middle), water temperature (purple, lower), and air temperature (green, lower) for the sampling season (2017). Meteorological variables were averaged over the catchment area, whereas water discharge and temperature were measured at the outlet (MR/ER). Shaded areas show the range of values of temperature within a day at the 4 stations (air, green) and at the outlet (water, purple), the range of flow at the outlet for discharge, and show the error of the snow covered area estimated from satellite data. Sampling days are indicated as black vertical lines with the corresponding day of the year on the lower axis.

Three percent (286) of ZOTUs had a taxonomic assignment to genus level (Supplementary File S6), of which 14 ZOTUs significantly associated with the main channel, 3 with springs, and 26 with tributaries according to the indicator analysis (Table S9). 17 ZOTUs were associated with tributaries and main channel sites. Multiple ZOTUs assigned to the genus of *Diamesa* were associated to the main channel or both main channel and tributaries, confirming traditional observations (Milner and
5 Petts, 1994). It is a common challenge of eDNA metabarcoding for the COI gene that only a low proportion of ZOTUs have an associated taxonomic name (here 3%, representing only 5% of the reads), which is largely resulting from lacking reference data (Weigand et al., 2019). Thus, all further analysis were done at the level of ZOTUs, irrespective if they had a taxonomic name assigned to it or not, which is suggested to be the future avenue of metabarcoding (Apothéloz-Perret-Gentil et al., 2017; Cordier et al., 2018).

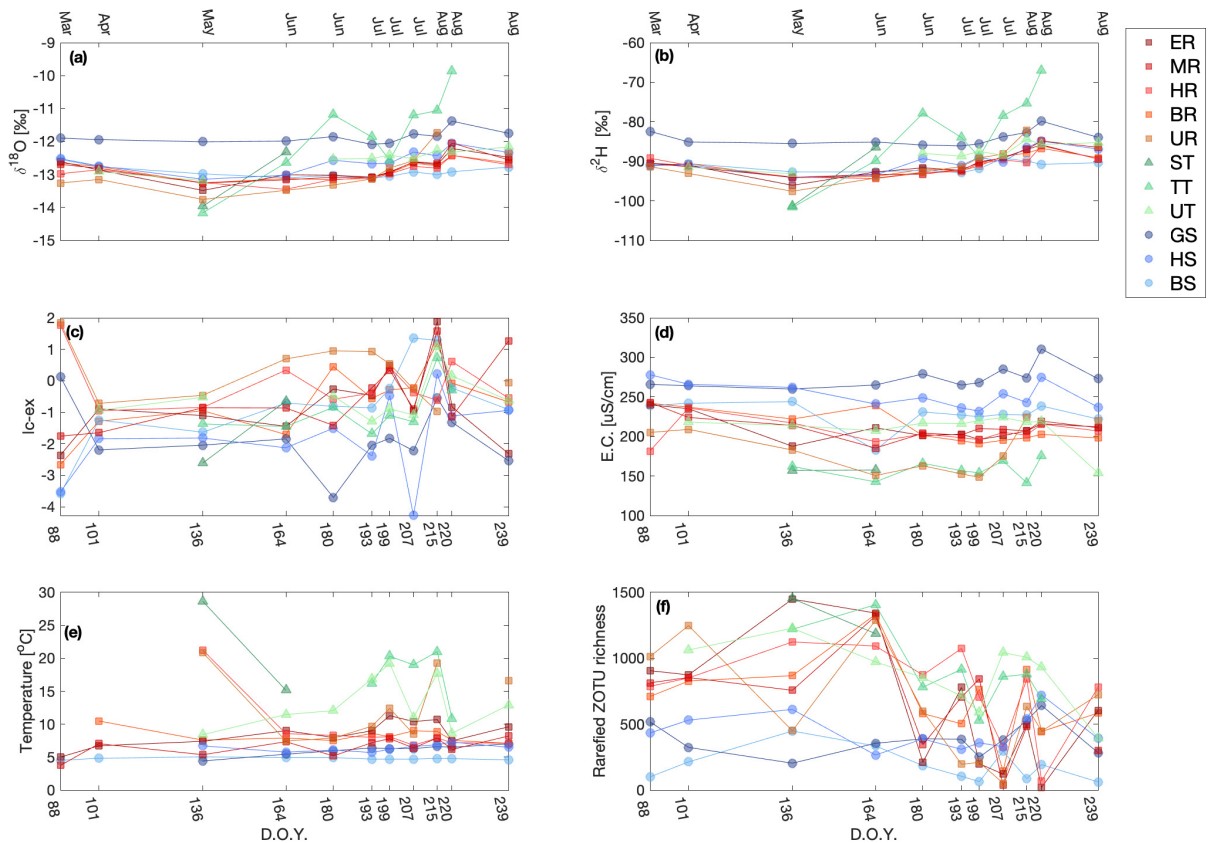

**Figure 3.** Variation in $\delta^{18}O$ (a), $\delta^2H$ (b), lc-ex (c), E.C. (d), temperature (e), and rarefied ZOTU richness (f) for the individual sampling sites over the field season (given as day of the year). The sampling sites in the legend are clustered based on the water source and within the cluster ordered from low to high elevation. The figure legend corresponds exactly to that in Fig. 1 and the x-axis corresponds exactly to that in Fig. 2.

### 3.3 Influence of hydrologic variability on eDNA diversity

The link between environmental variability in the different water sources and detected richness was pronounced: Both change of discharge and E.C. affected detected richness (Fig. 5a respectively 5b); and the best model included the interaction of the fixed effects for both the model with *dq/dt* and the model with E.C. (Table S10). The first model, including *dq/dt*, revealed that the fixed effects and their interaction were significant (Table 1, see Table S11 for results of random effects). However, water source is a factorial variable and the individual sources showed not the same response: The intercept (-0.637) and the slope (-2.055) for the spring samples are negative compared to the main channel, indicating an inverse effect of *dq/dt* on the richness detected in springs. The slope for tributary sites is smaller compared to the main channel sites but still resulting in a





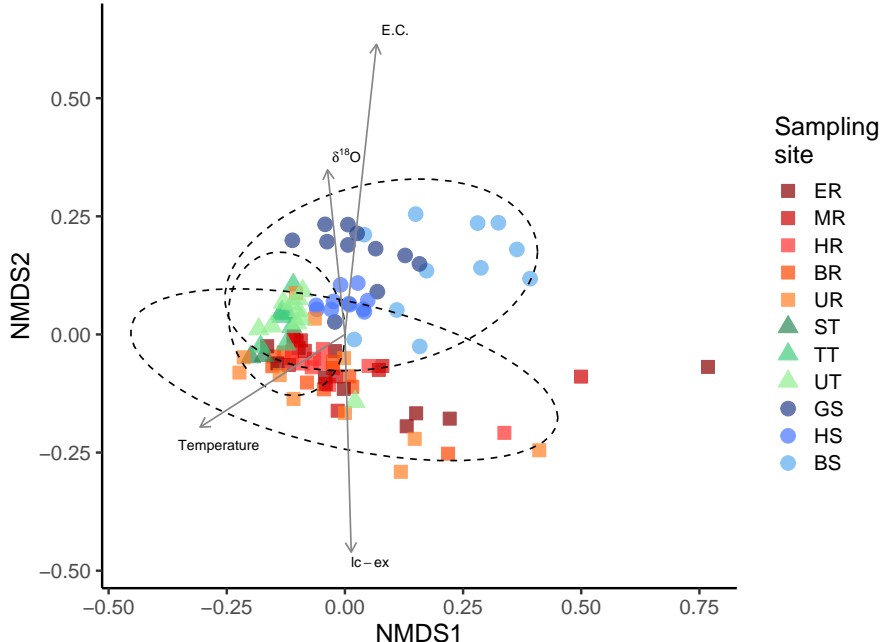

**Figure 4.** Community composition in the main channel, tributaries, and springs. Non-metric multidimensional scaling analysis of communities based on UniFrac dissimilarities (stress = 0.132, indicating a fair representation of the data in the reduced dimensions). Ellipses represent a 95% confidence interval.

positive relationship of richness and *dq/dt*. The second model, including E.C. instead of *dq/dt* as a fixed effect, identified all fixed effects and their interaction to be significant, except the interaction of E.C. and tributary sites; at this level of the factorial variable, water source is not related to E.C.

### 3.4 Separation of upstream water contribution

5 Over the whole sampling campaign, 79.2% of ZOTUs found in tributaries were recovered in the main channel, compared to 77.0% found in springs and the main channel (Fig. 6). Over all ZOTUs, we found 1,374 ZOTUs (14.3%) that were detected only in spring sites but never tributaries and 3,221 ZOTUs (33.5%) that were exclusively assigned to tributary sites. The best generalized linear model for *dq/dt* included the interaction with the origin of the ZOTUs (Table S12). Only the contribution from ZOTUs originating in tributaries to the main channel had a significant interaction with *dq/dt*, while no significant interactions 10 were observed for ZOTUs originating from springs (Table 2, Fig. 6b). In contrast, the best model with E.C., which was more correlated with richness in springs, did not include an interaction term (Table 2, Fig. 6c).



**Table 1.** Results from the generalized linear mixed model fit by maximum likelihood (Laplace approximation) for fixed effects, asterisks stand for the interaction. Note that the water source is a factor with three levels (main channel, spring, and tributary).

| Model | Fixed effects | Estimate | Std. error | z value | P-value |
|-------|---------------|----------|------------|---------|---------|
| *dq/dt* | Main channel (intercept) | 6.407 | 0.093 | 68.888 | < 0.001 |
| | *dq/dt* | 1.791 | 0.026 | 69.582 | < 0.001 |
| | Spring | -0.637 | 0.152 | -4.192 | < 0.001 |
| | Tributary | 0.434 | 0.152 | 2.855 | 0.0043 |
| | *dq/dt*\*Spring | -2.055 | 0.053 | -38.485 | < 0.001 |
| | *dq/dt*\*Tributary | -1.128 | 0.048 | -23.668 | < 0.001 |
| E.C. | Main channel (intercept) | 6.569 | 0.090 | 73.092 | < 0.001 |
| | E.C. | 0.201 | 0.010 | 20.335 | < 0.001 |
| | Spring | -1.183 | 0.149 | -7.946 | < 0.001 |
| | Tributary | 0.549 | 0.149 | 3.705 | < 0.001 |
| | E.C.\*Spring | 0.156 | 0.024 | 6.517 | < 0.001 |
| | E.C.\*Tributary | -0.027 | 0.021 | -1.272 | 0.203 |

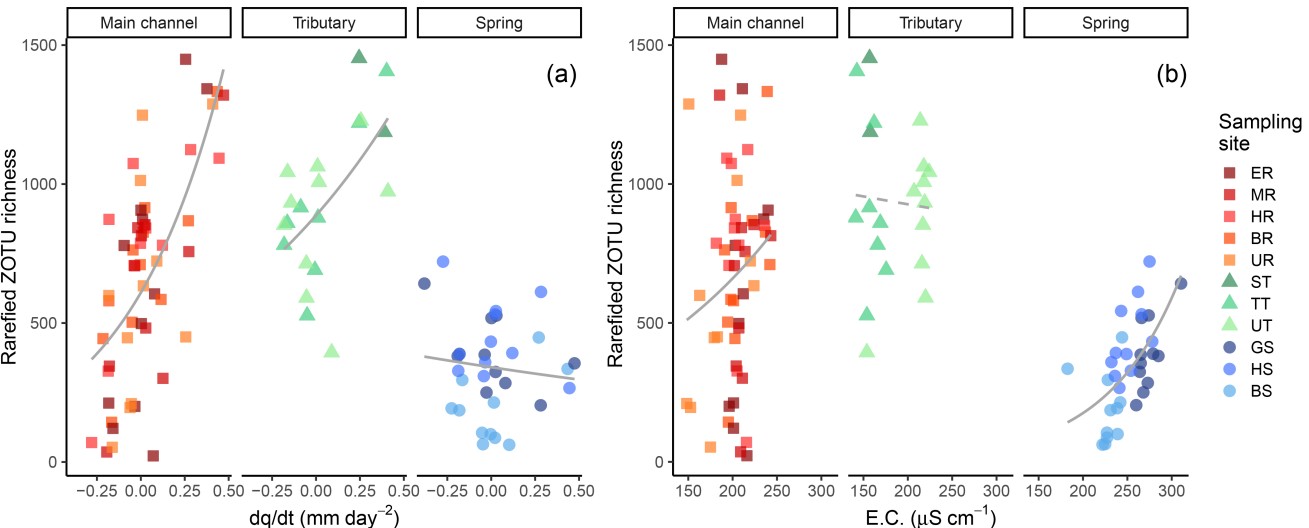

**Figure 5.** Rarefied ZOTU richness plotted against *dq/dt* (a) and E.C. (b). The sampling sites in the legend are clustered based on the water source and within the cluster ordered from low to high elevation. Solid lines indicate significant interactions of fixed effects and the dashed line indicates significant relationship of the fixed effects and the response variable only (without observed significant interaction). Smoothing lines are based on the GLM function.



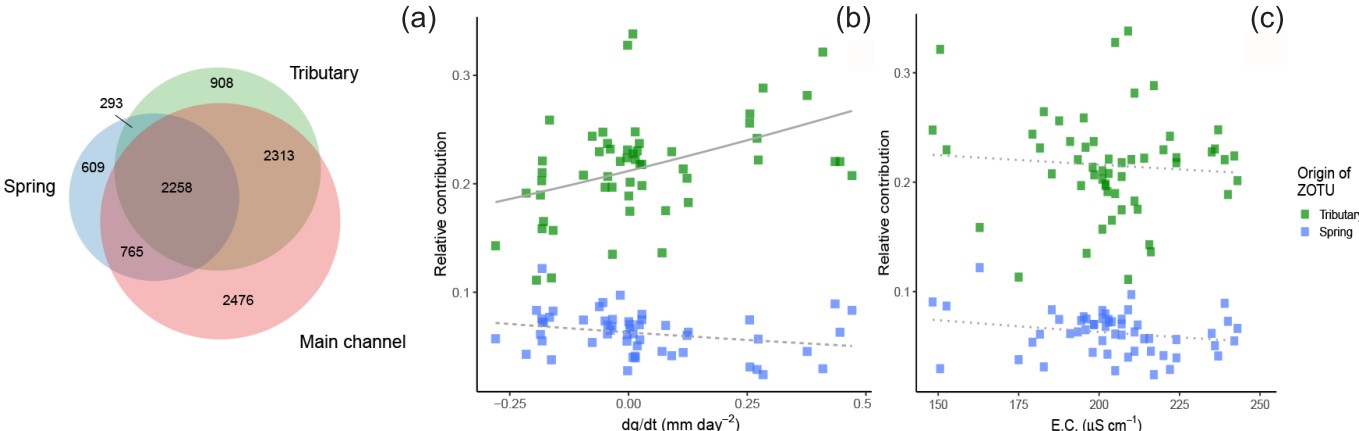

**Figure 6.** Venn diagram for the three sampled water sources, overlapping areas depict shared ZOTUs between water sources (a). Values indicate the numbers of ZOTUs in the given section and size of the circles are weighted by the number of ZOTUs. Relative contribution of ZOTUs from tributaries (green) or springs (blue) to samples in the main channel against *dq/dt* (b) and E.C. (c). The solid line indicates significant interactions of fixed effects, the dashed lines indicate significant relationships of both fixed effects and response variable only (without significant interaction) and the dotted line indicates only the significant relationship of one fixed effect. Smoothing lines are based on the GLM function.

**Table 2.** GLM results for response variables based on a t-test, asterisks stand for the interaction. Note that the origin of the ZOTU is a factor with two levels (spring and tributary).

| Model | Coefficients | Estimate | Std. Error | t value | P-value |
|---|---|---|---|---|---|
| *dq/dt* | Spring (intercept) | -2.700 | 0.052 | -51.548 | < 0.001 |
|  | *dq/dt* | -0.498 | 0.298 | -1.673 | 0.09724 |
|  | Tributary | 1.386 | 0.061 | 22.685 | < 0.001 |
|  | *dq/dt* *Tributary | 1.147 | 0.340 | 3.368 | 0.00106 |
| E.C. | Spring (intercept) | -2.377 | 0.270 | -8.790 | < 0.001 |
|  | E.C. | -0.001 | 0.001 | -1.255 | 0.212 |
|  | Tributary | 1.416 | 0.064 | 22.063 | < 0.001 |

## 4   Discussion

Our results highlight that eDNA can provide additional insights regarding the water type and origin when accompanied by observations of naturally-occurring physio-chemical tracers. Additionally, eDNA delivers information about time points of high connectivity between the sampling sites in a hydrologic highly diverse Alpine system. Thereby, it complements the classical

5   observation of stable isotopes of water, water temperature, and E.C. The measure of taxonomic richness assessed by eDNA showed highest values in tributaries, especially compared to springs, and showed less variation over time in tributaries than the





main channel. Below, we further elaborate on what we learn about the hydrology of the study site from eDNA observations and on the relationship between eDNA and the two physical variables that were most strongly associated with it, namely the change in discharge $dq/dt$ and E.C. Overall, in response to our own specific questions and hypotheses, we found that the eDNA signal not only provided unique biological information, but also partially discriminated water types, reflected hydrologic variability, and was determined by upstream contributions.

## 4.1 Differentiation of water type by eDNA

Our analysis showed that the eDNA composition of the three water types was indeed different, but not to a level that made them entirely distinct. In fact, we always expect a portion of the eDNA signal that is non-informative on the water types, and this overlap can be explained by either shared species compositions due to ecological connectivity between sites (Pringle, 2003; Bracken and Croke, 2007) and/or by transport of eDNA between hydrologically connected sites, i.e. downstream in the main channel or laterally through groundwater exchange with the hyporheic zone (Bracken et al., 2013). Even through a hydrologic lens, these three water types are not entirely distinct, for example groundwater contributes to the flow in the tributaries and main channel. The feasibility of using eDNA data to disentangle different hydrologic contributions in low-land streams, specifically inflows from waste-water treatment plants into natural streams, has recently been demonstrated (Mansfeldt et al., 2020). Thus, in a context of completely distinct water sources, a source tracking with DNA from microorganisms is possible, but may be more challenging in natural Alpine streams with overlapping signals (Wilhelm et al., 2013).

We identified E.C. and temperature as the two most important indicators for the community clustering over all samples. This corroborates previous, classical Alpine stream biodiversity work (e.g., Milner and Petts 1994; Ward 1994; Brown et al. 2003). Interestingly, phylogenetic community composition of the tributaries are between the main channel and the spring sites, represented in the intermediate position of the clustering in the NMDS. The transport of eDNA is not able to override these differences, otherwise we would expect the main channel to cluster between the tributary and the spring sites. By monitoring over a season, we identified timepoints of greater and lesser homogeneity among sites according to the eDNA community. This is consistent with the different hydrologic characteristics: while the main channel and tributaries resembled each other more (i.e., were more connected) on days with increased precipitation or snowmelt (e.g., DOY 136, 164, or 215), springs fed by groundwater were more stable over time and showed no reaction. eDNA provides promising new insights into the temporal evolution of the connectivity of the stream network system, a key to assessing dominant hydrologic flow paths.

Future studies might investigate how community composition could be exploited to enhance knowledge about ecosystem processes. However, our analysis was restricted to ZOTUs identity, as currently the assignment species names is challenging, likely because the database coverage for reference sequences is still limited (Weigand et al., 2019) and might be even lower for high alpine areas. While ZOTUs do not tell us anything about the identity of the organisms behind, they are sufficient to uncover shared vs. unique signals from different water types. The advantage of the ZOTU level is that the analysis is not restricted to a subset of organisms that are commonly used in aquatic ecology, or those that are covered in the databases, but are using an overall "fingerprint" of organisms.



### 4.2 Influence of hydrologic variability on eDNA diversity

#### 4.2.1 Change in discharge

Despite all variability, there was a positive relationship between the change in stream flow, *dq/dt*, and detected eDNA richness in the tributaries and main channel. High discharge events resulted from either rainfall or snowmelt, and cause expansion of
the surface channel network, or in other words, increased overland and subsurface flow (Chorley, 1978), a well-established consequence (Biswal and Marani, 2010; Mutzner et al., 2013). Likely, this expansion put water in contact with additional surface areas, likely resulting in the wash-in of terrestrial-origin eDNA, and causing an increase in the diversity signal. However, due to the lack of taxonomic information the exact origin of these ZOTUs (i.e., terrestrial origin or not) cannot completely be identified. Conversely, when stream flow decreased, corresponding to an overall contraction of the network, reduced overbank
flow, less exchange with the hyporheic zones and diminution of channel bank erosion, likely led to more characteristic DNA of the sampled water types in our record (Fig. 7). The tributaries can be considered as the *suppliers* of the ecohydrologic system, gathering genetic diversity when they contain more water and hydrologic connectivity increases. The richness at spring sites decreased with increasing *dq/dt*, perhaps because contributions from precipitation diluted the eDNA signal. This is further supported by the observed positive relationship between *dq/dt* and an increased number of ZOTUs originating from tributaries
but not from springs.

#### 4.2.2 Electrical conductivity

Detected richness in springs was relatively stable through the seasons (as were stable isotopes and temperature); we even observed a slight negative relationship between richness and *dq/dt*. This is not surprising, as the springs have a relatively constant flow and stable surface area. However, E.C. increased over the season in springs, indicating elevated cumulative
contact with subsurface substrate (Malard et al., 1999), which also positively correlated with richness. Our models suggest that, similarly to surface water, richness in springs could be increased by interaction with other habitats, in this case potentially sub-surface areas (indicated by the higher E.C.). However, we found a relatively small contribution of eDNA from springs to the main channel, at least during our sampling season, and no relationship between E.C. and the contribution of ZOTUs to the main channel from springs. The quantity of the contribution from springs to the main channel, in terms of both eDNA and
water, is most likely greater and thus more visible during winter base-flow conditions (Zhang et al., 2018), which were omitted from our study.

### 4.3 Separation of upstream water contribution

We hoped that eDNA would reveal where water flowed before emerging as surface water, such that it can be used to assess hydrologic connectivity. We tested this by comparing DNA from water types (tributaries and springs) with that of the main
channel. There was a reasonably high level of recovery in the main channel, 77-79%, in terms of ZOTUs, and a level, though minimal, of specificity to either springs or tributaries (14-33%, respectively). Potentially, the spatial scale matters, as several





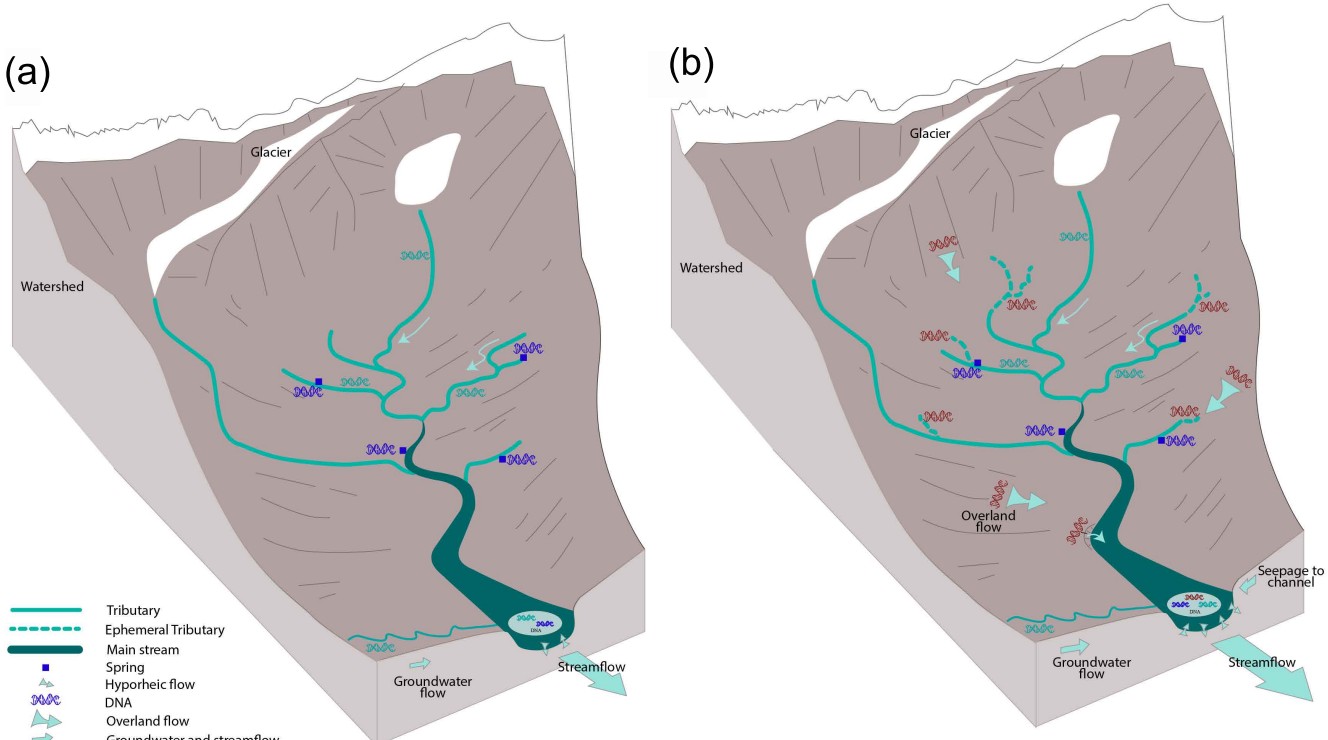

**Figure 7.** Conceptual diagram of relation between *dq/dt* and eDNA richness in the hydrologic system. When *dq/dt* is negative (a), the limited hydrologic connectivity and the limited connectivity between the terrestrial and aquatic habitat led to a segregated, water source-specific eDNA record; whereas when it is increasing (b), exchange is enhanced between the stream and the hyporheic and riparian zones through hyporheic connectivity, overland flow, and groundwater, which leads to higher diversity of suspended eDNA at the outlet.

studies have shown that eDNA can travel 10 to 100 km (e.g., Deiner and Altermatt 2014; Pont et al. 2018). The transport distances in our study system can be assumed to have minor influence on the degradation of eDNA, since they are comparatively small, and accordingly decay of eDNA can be neglected as a first approximation.

To calculate the hydrologic source contributions over time, we decomposed the water from the main channel to the two origins based on detected eDNA traces. This has previously be done using stable isotopes of water for example by Ohlanders et al. (2013) to identify the glacial melt contribution to discharge. In order to use eDNA for this calculation, we made the following assumptions, 1) that the percent of the total ZOTUs found in water from the main channel, which were also found in either springs or tributaries, reflect the water flow from those channel types, 2) that the rarefied number of ZOTUs equally





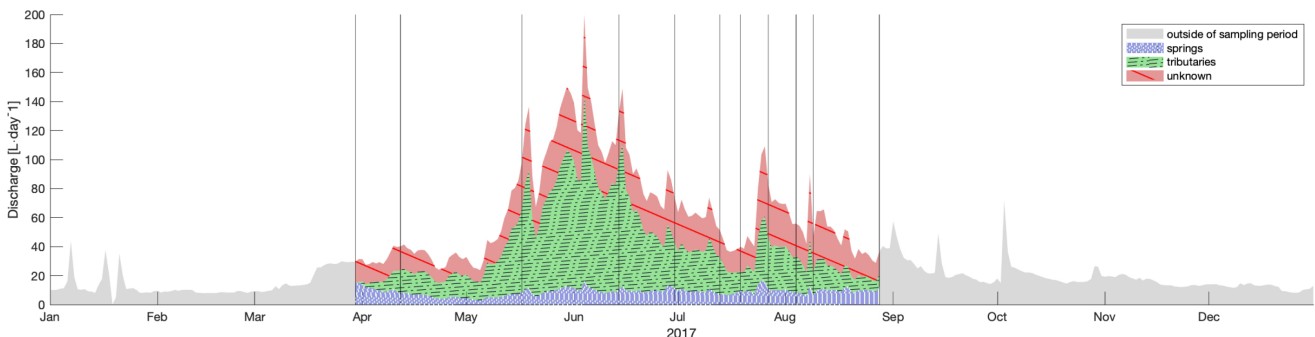

**Figure 8.** Discharge at outlet decomposed into water origin based on sampled eDNA on 11 days. Contribution of volume from springs is shown in blue, from tributaries in green, and from unknown sources in red. The annual hydrograph is shown for 2017, however volume outside of the sampling period is not computed and is shown in grey. Vertical lines indicate days when eDNA was sampled.

represented present ZOTUs as opposed to a bias introduced by the rarefaction step, 3) that each of those ZOTUs is composed of a similar number of species with a normally distributed amount of shed eDNA, and 4) that we can interpolate linearly to compute contributions of water types between sampled days (intervals ranging from 5-35 days). Because the volume of sampled water was consistent, we can estimate that there was a total contribution of 14% and 50% from springs and tributaries,

respectively over the sampled season (Fig. 8). An annual total of almost 13.5 m$^3$ flowed thorough the outlet in 2017, of which 9.9 m$^3$ flowed during the eDNA sampling period and of that, we estimate that 1.4 m$^3$ and 5 m$^3$ were from the sampled or similar tributaries or springs, respectively. The remaining 3.6 m$^3$ must be from other, not sampled sources. Our decomposition of the hydrograph is based on eDNA flow between sources and the main channel. Most hydrologic models base hydrograph decomposition on source area, runoff process, or input magnitudes without a validation method (Weiler et al., 2018). Our

thought experiment regarding an eDNA basis for hydrograph decomposition can now be compared to models with the same goal.

## 4.4   Recommendations and future steps

eDNA appears to offer high potential to not only provide insights into ecological diversity but also to inform interpretations of hydrological flow path connectivity. Future studies using eDNA as a supplemental indicator of hydrologic processes would

benefit from more targeted sampling campaigns in terms of hydrologic conditions, including a systematic sampling of discharge recession periods or sampling before and after high flow events generated by different hydrologic processes (e.g., high intensity precipitation events, snow melt-induced floods, glacial melt). Although the construction of a complete species inventory was not feasible with our data, with the implementation of the indicator analysis, we did a first step towards the identification of key biota in natural occurring eDNA in three characteristic Alpine water types. Future studies may implement the collection of

water from additional origins, such as glaciers, permafrost, hyporheic zones, or swift channels, to identify characteristic biota that could be used as tracers and their respective hydrologic contributions.





In addition, our results expose the complexity of using a metabarcoding approach for riverine biodiversity assessments on a broad taxonomic scale; it is not possible to differentiate between confounding hydrological processes and cycles with opposite effects on diversity. We recommend that future assessments of species richness in Alpine stream networks base their sampling on two aspects, depending on their goals. If the goal is to detect the maximum diversity of a targeted catchment,

then the detected diversity can be maximized with a single sampling campaign just after a rain or snowmelt event when the homogenization of the eDNA is highest, and we do not see differences in the species composition according to water type. In contrast, if the goal is to detect habitat-specific diversity or rare species, then sampling after an event resulting in high $dq/dt$ would not be recommended. In this case, the various habitats, or water sources, should be sampled when the environment is the most stable. Similarly, it is possible that E.C. could be used to guide sampling decisions to target specific flow conditions.

Finally, future sampling campaigns should have a clear design to answer specific questions about connectivity i.e. how the sampled eDNA is reflecting the hydrologic and geomorphic, versus ecologic connectivity in space and time.

## 5   Conclusions

Our study is a first evaluation of the spatial and temporal heterogeneity of eDNA diversity in an Alpine stream system, and how it relates to observed naturally-occurring hydrologic tracers. The genetic data provided a new axis of information regarding

the biological diversity, which in some locations and at certain moments in time corroborates the understanding of hydrologic processes gained from observation of electrical conductivity, stable isotopes of water, or water temperature, while also bringing new insights regarding the connectivity of water flow. Most importantly, we identified that some genetic signals are associated with specific water types encompassing the characteristic water flow paths and sources of this catchment, especially springs. Furthermore, the genetic signals demonstrate seasonal patterns according to water type. Consequently, eDNA may have a

high potential as an additional variable to be included in hydrologic studies because it not only provides data that informs understanding of hydrologic processes but also provides insight into the ecological diversity and habitat characteristics. Combined hydro-ecological knowledge is essential for a coherent and complete picture of Alpine aquatic systems that can inform understanding and management of the physical and living environment.





*Data availability.* Sequencing data will be publicly available on European Nucleotide Archive and all other data (water temperature, E.C., stable isotopes, etc.) are published on Zenodo (http://doi.org/10.5281/zenodo.3515062).

*Author contributions.* EM, FA, AS, AL, BS, and NC conceived the ideas and designed methodology; EM, AS, JCW, AL, and NC collected the data; EM and NC analyzed the data; EM and NC led the writing of the manuscript; AL contributed Fig. 7. All authors contributed

critically to the drafts and gave final approval for publication. FA and BS secured funding.

*Competing interests.* Author BS is a member of the editorial board of the journal, but otherwise there are no competing interests present that the authors are aware of.

*Acknowledgements.* Data analyzed in this paper were generated in collaboration with the Genetic Diversity Centre (GDC), ETH Zurich. Field work and infrastructure was supported by the Institute of Earth Surface Dynamics at the University of Lausanne, in particular by

10 all members of the catchment hydrology group for field and laboratory work, and by Dr. Torsten Vennemann and Dr. Jasquelin Peña for laboratory equipment and space. We would like to thank Samuel Hürlemann and Marco Thali for help in the lab. We thank Scott Blankenship and an anonymous reviewer for their significant and constructive feedback on earlier manuscript versions.

*Financial Support.* Funding is from the Swiss National Science Foundation Grants No PP00P3_179089, 31003A_173074 (to FA), PP00P2_157611 (to BS), the Velux Foundation, and the Chuard Schmid Foundation (to BS).



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
