# Peer review of "Environmental DNA simultaneously informs hydrological and biodiversity characterization of an Alpine catchment"

_Hydrology and Earth System Sciences, 2020_

## Referee Comment (RC1) · Anonymous Referee #1 · 10 Nov 2020

This manuscript presents genetic data collected in an Alpine catchment at various hydrologically relevant spatially distributed locations and as a function of time. The idea to link aquatic diversity with hydrologic processes is interesting. Authors present a coherent, innovative, and I think rather labour intensive work, which starts answering questions related to hydrologic connectivity of sources and eDNA diversity. I am not an expert in eDNA, but to me, the enormous variability seems an issue. I also see that authors recognize this. My main (minor) problem with the approach followed is that authors seem to lose sight of eDNA mass. Perhaps eDNA diversity weighed for mass could have been beneficial in order to reduce the diversity somewhat, and to focus more on a subset of most important ZOTUs or something like that. I understand that

the various steps in determining DNA sequences prevent working quantitative. In the future work section they could perhaps devote some attention to this aspect.

Two micro issues:

On page 2, Line 31 sodium. I don't get it why sodium all of a sudden is so important here. Usually chloride is more important as this behaves conservative in groundwater.

Page 3 L18: stream): the opening parenthesis is missing.

————————————————————

---

## Short Comment (SC1) · 13 Nov 2020

Dear Reviewer

We are very thankful for the time and feedback you gave us. We will carefully revise the manuscript once we receive the additional revisions and the editor's response.

Yours Sincerely, Elvira Mächler

---

## Referee Comment (RC2) · Anonymous Referee #2 · 10 Dec 2020

This is a very "dense" paper where the authors propose DNA-based indicators that simultaneously include information about the hydrological and biological features of the stream network of Alpine systems. The approach comes from the consideration that in these systems the high variability of physio-chemical properties and flow paths frequently corresponds to that observed in biological habitats. In these habitats, highly specialised organismal communities tend to develop according to the trophic status, which in turn is related to the source of the water "type". For example the three aquatic environments (tributaries, springs, and the main channel) are unique habitats each with corresponding eukaryotic communities. Thus, the drift of biological organisms are expected to have the potential to trace connectivity of the stream network. As microorganisms leave traces of their DNA in the environment, this DNA (environmental DNA - eDNA) may be used as a tracer to derive flow patterns in a watershed using hydrologic models. In their paper, the authors evaluated the possibility of using eDNA in hydrologic assessments of an Alpine system and, contextually, to gain insights on where and when to sample eDNA in river networks for assessments of biological diversity. To do that, a very intensive monitoring campaign was set up in an Alpine catchment in Switzerland, where they monitored simultaneously eDNA, electrical conductivity, water temperature, stable isotope ratios of the water, as well as discharge at the catchment outlet and meteorological parameters at four stations distributed across the catchment at different a.s.l... The authors used so-called ZOTUs (clusters of very similar DNA sequences) as a rough proxy for a species present in different aquatic systems and thus indicating different water origins. At the same time, the authors also used the derivative of the discharge at the outlet, dq/dt, as a proxy for stream network recession and expansion. At the end, they discussed the relationships among the different indicators considered

General comments The manuscript is very well structured. The introduction of the paper illustrates clearly the rationale and the objectives of the work. It provides a wide and exhaustive literature review about the approach used. The figures depict clearly the experimental data and, in general, the Materials and Methods are well explained. The number of techniques and methodological analyses used requires multidisciplinary skills to be correctly interpreted. I am not a biologist and the techniques to analyse the DNA should be revised by a reviewer with specific skills As for the approach and the interpretation of the results, based on my reading of the manuscript, I identified some strength and weakness points. The strengths mostly lie in the multidisciplinary approach on one side and, on the other side, in the number and quality of measurements the authors did in terms of eDNA, electrical conductivity, water temperature, stable isotope ratios of the water, discharge at the catchment outlet and meteorological parameters. Quite interesting is the use of the eDNA to identify (at least qualitatively) times of greater and lesser interconnection among water in different sites in the stream network,

so that the main channel and tributaries resembled each other more (i.e., were more connected) on days with increased precipitation or snowmelt. The mechanism is quite clearly shown in the figure 7. Weaknesses are mostly related to the interpretation of the measurements and the relationships between eDNA and "type" of water as related to its origin. For example, In the figure 5 I am not able to see a clear relationship between ZOTU richness and EC in the case of the main channel and tributaries, while it is a bit clearer for spring. I see a reversed situation in the relationship with dq/dt, even if, also in this case, a clear relationship does not exist even for main channels and tributaries. In any case, most of the deductions the authors drew in the paper comes from a statistical analysis, which, at least in this specific case, can indicate something behind the observed behaviour but are not able "to see" the actual mechanisms inducing different DNA composition in the different water types in different times. In this sense, the deductions of the authors seems, to me, a bit speculative. Actually, the same authors stated: "Our analysis showed that the eDNA composition of the three water types was indeed different, but not to a level that made them entirely distinct. In fact, we always expect a portion of the eDNA signal that is non-informative on the water types, and this overlap can be explained by either shared species compositions due to ecological connectivity between sites and/or by transport of eDNA between hydrologically connected sites". Even the potentiality of using the eDNA to identify times of greater and lesser interconnection among water in different parts of the stream network seems mostly qualitative. From the results analysis, it seems clear that the eDNA cannot replace the classical indicators (stable isotopes of water, water temperature, and E.C.) to discriminate among different origin of the water in the network. And yet, the eDNA analysis can still be used to support the observations with physio-chemical tracers, which are themselves not so simple to interpret.

Specific remarks The first nine lines of the abstract should be moved to the Introduction section. In figure 4, the caption should indicate the meaning of NMDS1 and NMDS2
* * *
490, 2020.

---

## Short Comment (SC2) · 10 Dec 2020

Dear Reviewer

We are very grateful for your time and feedback. We will carefully revise the manuscript after the open review period closes and further instructions by the editor are given.

Yours Sincerely, Elvira Mächler
* * *

---

## Author Comment (AC1) · 21 Dec 2020

Thank you for taking the time to read and comment on our manuscript, we appreciate your helpful and constructive feedback. Please find below a point by point response (in *italics*) to your comments. We feel confident that a clarification as response is sufficient to address the comments of the reviewers. These clarifications could be transferred to the manuscript in a more condensed manner, which we would be happy to implement.

Sincerely,

The Authors

[Figure]

**Refree 1**

This manuscript presents genetic data collected in an Alpine catchment at various hydrologically relevant spatially distributed locations and as a function of time. The idea to link aquatic diversity with hydrologic processes is interesting. Authors present a coherent, innovative, and I think rather labour intensive work, which starts answering questions related to hydrologic connectivity of sources and eDNA diversity.

*> Thank you for appreciating our work and acknowledging the novelty of our approach.*

I am not an expert in eDNA, but to me, the enormous variability seems an issue. I also see that authors recognize this. My main (minor) problem with the approach followed is that authors seem to lose sight of eDNA mass. Perhaps eDNA diversity weighed for mass could have been beneficial in order to reduce the diversity somewhat, and to focus more on a subset of most important ZOTUs or something like that. I understand that the various steps in determining DNA sequences prevent working quantitative. In the future work section they could perhaps devote some attention to this aspect.

*> We see there would be a benefit to have an additional measure of eDNA concentration as an proxy for eDNA mass; Indeed it is currently highly debated whether quantitative measures can be obtained from an eDNA metabarcoding approach due to various biases introduced in the laboratory procedures. To accommodate for this, we equalized the eDNA concentration of each sample before sequencing in order to get similar numbers of reads per eDNA sample, which is a standard practice in this field and likely already leads to a reduced diversity. Based on what we learned with this approach, in a future sampling effort at this site, it might be feasible to target specific sequences known to be tracers for specific processes or sources and assess them quantitatively, but according to our current understanding this would have to happen site-by-site and would not be easily transferable. We would be happy to clarify this opportunity in our outlook for the future.*
Two micro issues:

On page 2, Line 31 sodium. I don't get it why sodium all of a sudden is so important here. Usually chloride is more important as this behaves conservative in groundwater.

*> Indeed, we should change this wording from sodium to include all ions and minerals that could be in the water and drive electrical conductivity.*

Page 3 L18: stream): the opening parenthesis is missing.

*> Thank you for pointing out this error. We will remove the parenthesis as it is a relic of an older version of the manuscript.*

––––––––––––––––––––––––––

---

## Author Comment (AC2) · 21 Dec 2020

Thank you for taking the time to read and comment on our manuscript, we appreciate your helpful and constructive feedback. Please find below a point by point response (in *italics*) to your comments. We feel confident that a clarification as response is sufficient to address the comments of the reviewers. These clarifications could be transferred to the manuscript in a more condensed manner, which we would be happy to implement.

Sincerely, the authors

[Figure]

**Refree 2**

This is a very "dense" paper where the authors propose DNA-based indicators that simultaneously include information about the hydrological and biological features of the stream network of Alpine systems. The approach comes from the consideration that in these systems the high variability of physio-chemical properties and flow paths frequently corresponds to that observed in biological habitats. In these habitats, highly specialised organismal communities tend to develop according to the trophic status, which in turn is related to the source of the water "type". For example the three aquatic environments (tributaries, springs, and the main channel) are unique habitats each with corresponding eukaryotic communities. Thus, the drift of biological organisms are expected to have the potential to trace connectivity of the stream network. As microorganisms leave traces of their DNA in the environment, this DNA (environmental DNA - eDNA) may be used as a tracer to derive flow patterns in a watershed using hydrologic models. In their paper, the authors evaluated the possibility of using eDNA in hydrologic assessments of an Alpine system and, contextually, to gain insights on where and when to sample eDNA in river networks for assessments of biological diversity. To do that, a very intensive monitoring campaign was set up in an Alpine catchment in Switzerland, where they monitored simultaneously eDNA, electrical conductivity, water temperature, stable isotope ratios of the water, as well as discharge at the catchment outlet and meteorological parameters at four stations distributed across the catchment at different a.s.l... The authors used so-called ZOTUs (clusters of very similar DNA sequences) as a rough proxy for a species present in different aquatic systems and thus indicating different water origins. At the same time, the authors also used the derivative of the discharge at the outlet, $dq/dt$, as a proxy for stream network recession and expansion. At the end, they discussed the relationships among the different indicators considered

General comments

The manuscript is very well structured. The introduction of the paper illustrates clearly the rationale and the objectives of the work. It provides a wide and exhaustive literature review about the approach used. The figures depict clearly the experimental data and, in general, the Materials and Methods are well explained. The number of techniques and methodological analyses used requires multidisciplinary skills to be correctly interpreted. I am not a biologist and the techniques to analyse the DNA should be revised by a reviewer with specific skills As for the approach and the interpretation of the results, based on my reading of the manuscript, I identified some strength and weakness points. The strengths mostly lie in the multidisciplinary approach on one side and, on the other side, in the number and quality of measurements the authors did in terms of eDNA, electrical conductivity, water temperature, stable isotope ratios of the water, discharge at the catchment outlet and meteorological parameters. Quite interesting is the use of the eDNA to identify (at least qualitatively) times of greater and lesser interconnection among water in different sites in the stream network, so that the main channel and tributaries resembled each other more (i.e., were more connected) on days with increased precipitation or snowmelt. The mechanism is quite clearly shown in the figure 7.

> *Thank you for appreciation of our work and for highlighting the interdisciplinary approach of our study.*

Weaknesses are mostly related to the interpretation of the measurements and the relationships between eDNA and "type" of water as related to its origin. For example, In the figure 5 I am not able to see a clear relationship between ZOTU richness and EC in the case of the main channel and tributaries, while it is a bit clearer for spring. I see a reversed situation in the relationship with dq/dt, even if, also in this case, a clear relationship does not exist even for main channels and tributaries.

> *We agree that some of the relationships are not obvious when looking at the figure.*

*It is important to note, though, that we are actually interpreting the data based on the statistical results presented in the Table 1, and not based on the figure itself. We tested for interaction (ZOTU richness = E.C. * water type), which allows us to identify if the intercept and slope of richness varies according to the water type, compared with an additive approach (ZOTU richness = E.C. + water type). An additive approach would only allow us to test whether the intercepts were different, while assuming the slopes were the same. To test for interaction was especially insightful in our case, because we identified different slopes (Figure 5a, dq/dt) for the each of the three water types (positive for main channel and tributary, but negative for spring). However, we only found a significant slope in terms of E.C. for the water types of main channel and the spring (Figure 5b). As you identified correctly, there is no significant interaction between E.C. and the water type of tributary (p-value of 0.203 in Table 1, i.e. the slope is not significant also indicated by the dashed line in the figure). In the case of the main channel, the interaction is significant, but perhaps adjusting the x-axis of E.C. for each water type would facilitate the interpretation, which we would be happy to do.*

In any case, most of the deductions the authors drew in the paper comes from a statistical analysis, which, at least in this specific case, can indicate something behind the observed behaviour but are not able "to see" the actual mechanisms inducing different DNA composition in the different water types in different times. In this sense, the deductions of the authors seems, to me, a bit speculative. Actually, the same authors stated: "Our analysis showed that the eDNA composition of the three water types was indeed different, but not to a level that made them entirely distinct. In fact, we always expect a portion of the eDNA signal that is non-informative on the water types, and this overlap can be explained by either shared species compositions due to ecological connectivity between sites and/or by transport of eDNA between hydrologically connected sites". Even the potentiality of using the eDNA to identify times of greater and lesser interconnection among water in different parts of the stream network seems mostly qualitative.

> *We used a non-metric multidimensional scaling method, thus we would argue that our approach of the NMDS analysis is quantitative, rather than speculative. But as you have pointed out, the stress of the NMDS was reasonable but certainly not high enough to lead to an excellent discrimination. Although it is outside of the scope of our study, a future study might want to sample from more distinct, unrelated sources such as the glacier directly, pore-water, snow, rainfall, rock ice or any stagnant terrestrial water pools. Furthermore, we want to highlight that we are interpreting a biological response which perhaps is not as cut and crisp as a binary response of a purely physical process might be.*

From the results analysis, it seems clear that the eDNA cannot replace the classical indicators (stable isotopes of water, water temperature, and E.C.) to discriminate among different origin of the water in the network. And yet, the eDNA analysis can still be used to support the observations with physio-chemical tracers, which are themselves not so simple to interpret.

> *We agree that based on our results, eDNA cannot replace any of these classical indicators. However, our results do demonstrate that eDNA provides a huge amount of information that complements existing indicators. The metabarcoding approach in particular offers a thorough snapshot into the biological communities inhabiting this environment which will be useful for a wide variety of goals. In addition, we believe that in the future, eDNA will help discriminate hydrological processes in a more nuanced fashion than is currently possible with physical indicators.*

Specific remarks
The first nine lines of the abstract should be moved to the Introduction section.

> *The first nine lines of the abstract do in fact summarize our current introduction quite well, with each sentence or clause introducing one paragraph of the introduction. We*

*believe that it is valuable to have this summary of the major background, problem, and opportunity in the abstract as that is the most accessed and read part of a paper, especially online. However, if the editor supports the reviewers view, we would be happy to reconfigure the introduction by including these lines and thus shorten the abstract to focus on our study and its results.*

In figure 4, the caption should indicate the meaning of NMDS1 and NMDS2.

*> We will add 'NMDS' in parentheses after non-metric multidimensional scaling in the figure legend and clarify that they stand for the dimension 1 (NMDS1) and dimension 2 (NMDS2).*

---

## Author Response (AR1)

**Dear Editor**

Thank you for taking the time to handle our manuscript. Please find below a point by point response (in *italics*) to the referees' comments. We give detailed responses to all of them. In virtually all cases, we correspondingly modified the manuscript. In one case, namely the suggestion to delete half of the abstract, we respectfully disagree and prefer to maintain the original version, as we feel this information provided in the abstract is needed for our manuscript to target an interdisciplinary readership. In all our responses, we made it clear what our action was. Please do no hesitate to let us know if further modifications are necessary. Furthermore, the genetic and environmental data are now in repositories and are publicly available. We have updated the corresponding access information in the 'Data Availability' section.

Sincerely,

The Authors

==============================================================================

**REVIEWER 1**

This manuscript presents genetic data collected in an Alpine catchment at various hydrologically relevant spatially distributed locations and as a function of time. The idea to link aquatic diversity with hydrologic processes is interesting. Authors present a coherent, innovative, and I think rather labour intensive work, which starts answering questions related to hydrologic connectivity of sources and eDNA diversity.

*> We thank the referee for the detailed and constructive feedback and evaluation of our work.*

I am not an expert in eDNA, but to me, the enormous variability seems an issue. I also see that authors recognize this. My main (minor) problem with the approach followed is that authors seem to lose sight of eDNA mass. Perhaps eDNA diversity weighed for mass could have been beneficial in order to reduce the diversity somewhat, and to focus more on a subset of most important ZOTUs or something like that. I understand that the various steps in determining DNA sequences prevent working quantitative. In the future work section they could perhaps devote some attention to this aspect.

*> We agree that measures of eDNA concentration as a proxy for eDNA mass would be an interesting and possible valuable avenue providing further information, but there are also technical challenges associated to it; Indeed, it is currently highly debated whether such quantitative measures can be obtained from an eDNA metabarcoding approach due to the influence of various steps in the laboratory procedures. To accommodate for this, we thus equalized the eDNA concentration of each sample before sequencing in order to get similar numbers of reads per eDNA sample (as already indicated in the supplementary material, section 'S3.2 Library preparation'), which is a standard practice in this field and possibly normalizes the diversity estimates too. We revisited the text and clarified that our approaches used reflect the current state of the art in the field, and any possible estimates on abundance and mass may be the goal of future studies.*

*> Specifically, based on what we learned with this approach, in a future sampling effort at this site, it might be feasible to target specific sequences known to be tracers for specific processes or sources and assess them quantitatively, but according to our current understanding this would have to happen site-by-site and would not be easily transferable.*

*> We are transparent about this variability and its origin in the second paragraph of section '4.3 Separation of upstream water contribution', where we discuss the assumptions required for the hydrologic source contributions overtime. We state that we assume that "each of these ZOTUS is composed of similar number of species with a normally distributed amount of shed eDNA." It is true that this mass balance calculation would be more precise if for each species we knew the initial quantity released, lost during transport, and sampled downstream. However, such information is not available for any species yet (not only in our study, but in general). We also added a new sentence proposing this idea to section '4.4 Recommendations and future steps'.*

**Two micro issues:**

On page 2, Line 31 sodium. I don't get it why sodium all of a sudden is so important here. Usually chloride is more important as this behaves conservative in groundwater.

*> Thank you. We now have rephrased it as: "For example, E.C. can relatively consistently discriminate snowmelt, rain, and potentially glacier melt, which all contain few solutes due to their little contact with rock and soil surfaces, from groundwater, which typically has much higher levels of solutes due to extended contact with surfaces (Williams 2006, Cochand 2019, Kobierska 2015).*

Page 3 L18: stream): the opening parenthesis is missing.

*> Thank you for pointing out this error. We removed the parenthesis as it is a relic of an older version of the manuscript.*
* * *
**REVIEWER 2**

This is a very "dense" paper where the authors propose DNA-based indicators that simultaneously include information about the hydrological and biological features of the stream network of Alpine systems. The approach comes from the consideration that in these systems the high variability of physio-chemical properties and flow paths frequently corresponds to that observed in biological habitats. In these habitats, highly specialised organismal communities tend to develop according to the trophic status, which in turn is related to the source of the water "type". For example the three aquatic environments (tributaries, springs, and the main channel) are unique habitats each with corresponding eukaryotic communities. Thus, the drift of biological organisms are expected to have the potential to trace connectivity of the stream network. As microorganisms leave traces of their DNA in the environment, this DNA (environmental DNA - eDNA) may be used as a tracer to derive flow patterns in a watershed using hydrologic models. In their paper, the authors evaluated the possibility of using eDNA in hydrologic assessments of an Alpine system and, contextually, to gain insights on where and when to sample eDNA in river networks for assessments of biological diversity. To do that, a very intensive monitoring campaign was set up in an Alpine catchment in

Switzerland, where they monitored simultaneously eDNA, electrical conductivity, water temperature, stable isotope ratios of the water, as well as discharge at the catchment outlet and meteorological parameters at four stations distributed across the catchment at different a.s.l... The authors used so-called ZOTUs (clusters of very similar DNA sequences) as a rough proxy for a species present in different aquatic systems and thus indicating different water origins. At the same time, the authors also used the derivative of the discharge at the outlet, dq/dt, as a proxy for stream network recession and expansion. At the end, they discussed the relationships among the different indicators considered

> *We thank the referee for the detailed and constructive feedback and evaluation of our work.*

**General comments**

The manuscript is very well structured. The introduction of the paper illustrates clearly the rationale and the objectives of the work. It provides a wide and exhaustive literature review about the approach used. The figures depict clearly the experimental data and, in general, the Materials and Methods are well explained. The number of techniques and methodological analyses used requires multidisciplinary skills to be correctly interpreted. I am not a biologist and the techniques to analyse the DNA should be revised by a reviewer with specific skills As for the approach and the interpretation of the results, based on my reading of the manuscript, I identified some strength and weakness points. The strengths mostly lie in the multidisciplinary approach on one side and, on the other side, in the number and quality of measurements the authors did in terms of eDNA, electrical conductivity, water temperature, stable isotope ratios of the water, discharge at the catchment outlet and meteorological parameters. Quite interesting is the use of the eDNA to identify (at least qualitatively) times of greater and lesser interconnection among water in different sites in the stream network, so that the main channel and tributaries resembled each other more (i.e., were more connected) on days with increased precipitation or snowmelt. The mechanism is quite clearly shown in the figure 7.

> *We thank the referee for the appreciation of the interdisciplinary nature of our work. This is indeed what we aimed for, and we have revisited the text and ensure that it is well-balanced for this readership.*

Weaknesses are mostly related to the interpretation of the measurements and the relationships between eDNA and "type" of water as related to its origin. For example, In the figure 5 I am not able to see a clear relationship between ZOTU richness and EC in the case of the main channel and tributaries, while it is a bit clearer for spring. I see a reversed situation in the relationship with dq/dt, even if, also in this case, a clear relationship does not exist even for main channels and tributaries.

> *This is a good point. We agree that some of the relationships are not obvious at first sight when looking at the figure, which is also partly due to the complex and interacting effects of the factors studied (even though not significant). It is important to note, though, that we are actually interpreting the data based on the statistical results presented in the Table 1, and not based on the figure itself. We tested for interaction (ZOTU richness = E.C. \* water type), which allows us to identify if the intercept and slope of richness varies according to the water type, compared with an additive approach (ZOTU richness = E.C. + water type). An additive approach would only allow us to test whether the intercepts were different, while assuming the slopes were the same. We added more*

*detailed explanations for this reasoning to the manuscript (section '2.4.2 Influence of hydrologic variability on eDNA diversity'). To test for interaction was especially insightful in our case, because we identified different slopes (Figure 5a, dq/dt) for the each of the three water types (positive for main channel and tributary, but negative for spring). It is a general issue of complex, possibly interacting factors having non-trivial effects (and thus sometimes not obvious at first glance); we revisited the figure and its caption to clarify.*

*> Importantly, we only found a significant slope in terms of E.C. for the water types of main channel and the spring. As the referee identified correctly, there is no significant interaction between E.C. and the water type of tributary (p-value of 0.203 in Table 1, i.e. the slope is not significant also indicated by the dashed line in the figure), which we stated already in the manuscript beforehand. We now more clearly indicate the value for the slopes estimated in Table 1, to facilitate interpretation of the table too. In the case of the main channel though, the interaction is significant. For Figure 5b, we adjusted the scale of the x-axis for each water type to facilitate the interpretation and added a note to the legend.*

In any case, most of the deductions the authors drew in the paper comes from a statistical analysis, which, at least in this specific case, can indicate something behind the observed behaviour but are not able "to see" the actual mechanisms inducing different DNA composition in the different water types in different times. In this sense, the deductions of the authors seems, to me, a bit speculative. Actually, the same authors stated: "Our analysis showed that the eDNA composition of the three water types was indeed different, but not to a level that made them entirely distinct. In fact, we always expect a portion of the eDNA signal that is non-informative on the water types, and this overlap can be explained by either shared species compositions due to ecological connectivity between sites and/or by transport of eDNA between hydrologically connected sites". Even the potentiality of using the eDNA to identify times of greater and lesser interconnection among water in different parts of the stream network seems mostly qualitative.

*> Indeed, all of our conclusions and deductions made in the manuscript are statistically supported and corroborated. We double-checked that this is clear at all places. Also, all of our analyses are quantitative, and not qualitatively only. Specifically, we used a non-metric multidimensional scaling method (NMDS), which is a broadly-used quantitative method. We ensured that there are no speculative claims made, and that we always clarify which statements are statistically corroborated and quantitative. As the referee correctly points out, the stress of the NMDS was reasonable but certainly not high enough to lead to complete discrimination (which we had already stated in the manuscript (section '3.2 Differentiation of water type by eDNA')). Thus, the discrimination is not absolute. Although it is beyond the scope of our study, a future study might want to sample from more distinct, unrelated sources such as the glacier directly, pore-water, snow, rainfall, rock ice or any stagnant terrestrial water pools to increase understanding of their respective hydrologic contributions. We modified a sentence from the previous version of the manuscript, which now more clearly indicates the benefit of such an additional sampling (section '4.4 Recommendation and future steps').*

From the results analysis, it seems clear that the eDNA cannot replace the classical indicators (stable isotopes of water, water temperature, and E.C.) to discriminate among different origin of the water

in the network. And yet, the eDNA analysis can still be used to support the observations with physio-chemical tracers, which are themselves not so simple to interpret.

*> We completely agree with this statement. We also do not want to imply that eDNA should replace any of these classical indicators and we now better clarify this in our conclusion section. Importantly, however, our results demonstrate that eDNA provides highly complementary information to existing indicators. The metabarcoding approach in particular offers a thorough snapshot into the biological communities inhabiting this environment, which will be useful for a wide variety of goals. Thus, while eDNA can complement some of the classic hydrologic indicators, it also gives novel information that has not been available by the physio-chemical measures only. Overall, we believe that eDNA will help discriminate hydrologic processes in a more nuanced fashion than is currently possible with physical indicators, in the future. We now clarified this aspect throughout the manuscript.*

**Specific remarks**

The first nine lines of the abstract should be moved to the Introduction section.

*> We respectfully disagree, and would like to keep these lines in the abstract. The first nine lines of the abstract are summarizing key information, with each sentence or clause summarizing one paragraph of the introduction. We believe that it is valuable to have this summary of the major background, problem, and opportunity in the abstract as that is the most accessed and read part of a paper, especially online. Also, we feel our paper targeting a highly interdisciplinary readership must have a sufficiently detailed abstract. We therefore decided to leave the text in the abstract as it was currently written, also given that neither the editor nor the first reviewer commented on this aspect.*

In figure 4, the caption should indicate the meaning of NMDS1 and NMDS2 4

*> We added 'NMDS' in parentheses after non-metric multidimensional scaling in the figure legend and clarified that they stand for the dimension 1 (NMDS1) and dimension 2 (NMDS2).*